# Effectiveness of Expanded Instantaneous Input Dynamic Range Adjustment on Speech Perception

**DOI:** 10.3390/jpm12111860

**Published:** 2022-11-07

**Authors:** Toru Sonoyama, Takashi Ishino, Takashi Oda, Yuichiro Horibe, Nobuyuki Chikuie, Takashi Kono, Takayuki Taruya, Takao Hamamoto, Tsutomu Ueda, Sachio Takeno

**Affiliations:** Department of Otorhinolaryngology, Head and Neck Surgery, Graduate School of Biomedical Sciences, Hiroshima University, Hiroshima 734-8551, Japan

**Keywords:** cochlear implant, speech perception, loud conversation, wide IIDR, noisy environment

## Abstract

Instantaneous input dynamic range (IIDR), as defined by Cochlear Ltd. (Sydney, Australia), refers to the acoustic level of short-term input dynamic range (IDR). Our aim was to evaluate the efficacy of expanding IIDR to improve speech understanding. We enrolled 11 unilateral Cochlear Ltd. patients with post-lingual hearing loss. The two types of IIDR settings (T-SPL/C-SPL of 25/65 dB (default IIDR) and 25/80 dB (wide IIDR)) were blindly assigned, and only one IIDR setting selected according to their preference was used for at least three months. Each IIDR group was evaluated with both default and wide IIDR conditions using the recorded word and sentence test materials of the Japanese CD speech discrimination scoring system (CI-2004 test) in quiet and noise with a signal-to-noise ratio (SNR) of +10 dB, presented at 65/80 dB SPL. Wide IIDR significantly improved speech perception in all tests, except for sentences in quiet conditions at a presentation level of 65 dB. Improvements during loud conversations in noisy environments were obtained without any adaptation period. Wide IIDR should become a new individual configuration setting method in Cochlear Ltd. devices to improve hearing in loud conversations and noisy environments.

## 1. Introduction

The electrical dynamic range (EDR), typically ranging from 6–15 dB SPL, is the range between the minimum current level, as the electrical stimulation threshold level (T level, as defined by Cochlear Ltd., Sydney, Australia), and the maximum current level, as the electrical stimulation loud-but-comfortable level (C level, as defined by Cochlear Ltd.) [1]. The acoustic dynamic range in people with normal hearing ranges from thresholds near 0 dB sound pressure level (SPL) to maximum tolerable levels of approximately 100–120 dB SPL, and the mean dynamic range of conversational speech ranges approximately 40–55 dB SPL, including intra- and inter-talker effects [2,3]. A microphone’s acoustic dynamic range, approximately 90 dB SPL, covering the mean dynamic range of conversational speech, is converted into the EDR using high compression. The range of adjusted acoustic sound levels mapped onto the EDR is called the input dynamic range (IDR). The instantaneous input dynamic range (IIDR), as defined by Cochlear Ltd., is the short-term IDR with a front-end compression limiter applied without Autosensitivity Control (ASC, as defined by Cochlear Ltd.), a fast-acting compression circuit for listening to soft speech (Whisper, as defined by Cochlear Ltd.), and using adaptive dynamic range optimization (ADRO, as defined by Cochlear Ltd.). IDR and IIDR are different terminologies, but both encompass the range of input acoustic signals.

All cochlear implant (CI) manufacturers set the default IDR/IIDR setting with a focus on an optimal auditory percept. Each CI manufacturer, except Neurelec, opts to maximize the intensity resolution using automatic gain control (AGC) at any given environmental noise level via setting the IDR/IIDR of 40 dB to between the T level (T-SPL) of 25 dB SPL and the input sound pressure level at the C level (C-SPL) of 65 dB SPL at the default setting (Custom sound^®^ v5.0, or later version) in Cochlear Ltd., 55 dB with a roving sound window of 25–100 dB in MED-El, and 60 dB at the default setting in Advanced Bionics, Valencia, CA, USA (SoundWave^®^ v3.1, or later versions). However, Neurelec opt to cover the diverse listening situations in daily life with a fixed IDR of 75 dB without the application of AGC [4]. In terms of the concept underlying IDR/IIDR settings in each CI manufacturer, Cochlear Ltd. (Sydney, NSW, Australia) devices focus on intensity resolution more than other CI devices do by applying a relatively narrow IDR/IIDR to fit the mean dynamic range in conversational speech. By contrast, the MED-EL, Advanced Bionics, and Oticon medical CI systems can process acoustic sound of 65–85 dB SPL without the application of infinite compression in the default setting by applying a relatively wide IDR/IIDR [4,5,6]. 

As the concepts underlying IDR/IIDR settings in each CI manufacturer are widely distributed, the effectiveness of IDR/IIDR expansion (wide IDR/IIDR) has long been discussed. The increase in IDR from 30 to 40, 50, and 60 dB in Advanced Bionics, LLC (Valencia, CA, USA) devices showed significant improvement in the AzBio sentence test presented at 57- and 77-dB SPL in quiet and 77 dB SPL in noise [7]. The IDRs of 65 and 80 dB in Advanced Bionics, LLC (Valencia, CA, USA) devices showed significant improvement in the consonant-nucleus-consonant (CNC) word test presented at 50 dB SPL compared with an IDR of 50 dB, and no significant differences between the three IDRs in the City University of New York (CUNY) sentence test in noise presented at 65 dB SPL [8]. In Cochlear Ltd. devices, an IIDR of 40 dB showed significantly higher scores for words (50 dB SPL) and sentences in background babble (65 dB SPL) and significantly lower sound field threshold levels of 40 dB compared to an IIDR of 30 dB [9]. IIDRs of 46 and 56 dB yielded significantly higher scores on the CNC word test presented at 45- and 55-dB SPL compared with an IIDR of 30 dB, and no significant differences between the three IIDRs for the CUNY sentence test in noise presented at 65 dB SPL [10]. These results revealed that speech perception, especially in soft sounds such as 50 dB SPL, can be improved by applying a wide IIDR. 

However, these studies did not report the effect of wide IIDR in speech perception in loud sounds such as 80 dB SPL. The C level in the 1-kHz channel is achieved at approximately 64 dB SPL for a speech-like signal at the default sensitivity setting of 12 [10], and excessive envelope levels exceeding 65 dB SPL as a temporal signal peak threshold are compressed by a fast-acting front-end compression limiter with infinite compression (Freedom/CP810/CP910/CP920/CP1000) and medium-acting AGC (CP810/CP910/CP920/CP1000). The fast-acting front-end compression limiter exhibits three detrimental effects at high presentation intensity exceeding the C-SPL. First, it distorts the spectral profile by creating distorted spectral signal peaks limited to the C-SPL, which potentially degrade vowel perception by modulating the formant frequencies at the attack time [11]. Second, it changes the modulation of temporal amplitude in the signal peak of the envelope at the attack time, degrading speech intelligibility [11]. Third, it reduces the signal-to-noise ratio (SNR) by compressing the peak amplitude in contrast to the background noise at the attack time [11]. These drawbacks may lead to a decrease in speech perception ability at high presentation intensity compared to normal to soft presentation intensity. It was already demonstrated that spectral profile distortion significantly decreased the perception of sentences in quiet conditions at 89 dB SPL and noisy conditions at over 80 dB SPL when changing the AGC system from a front-end limiter to an envelope profile limiter in Cochlear Ltd. devices [11]. 

In this article, we present the effectiveness of IIDR expansion when increasing C-SPL from 65 to 80 dB (wide IIDR) at normal and high presentation intensity in quiet and noisy environments by avoiding compression with AGC and pursuing an individualized IIDR adjustment method for Cochlear Ltd. patients, who often hear a loud voice rather than a soft voice. We believe that the wide IIDR setting may assist both patients with severe to profound hearing loss, who often hear a loud voice of around 65–80 dB from people accustomed to speaking loudly in their communication with the patients, and audiologists in further optimizing the individual fitting process.

## 2. Materials and Methods

### 2.1. Subjects

In this prospective study, we examined 11 adult unilateral CI users with bilateral post-lingual and profound sensorineural hearing loss (Table 1). This study design was planned under the consideration of cortical plasticity following cochlear implantation, because “auditory performance in CI users is seen to improve over the first 3.5 years following implantation” [12]; “the ability to repeatedly measure cortical plasticity both prior to and during this initial period of auditory recovery may be crucial in order to fully understand the role of the cortex in variable CI outcomes,” and “it proves notoriously challenging to obtain measures of cortical function repeatedly, and safely, over a short time period in CI users due to various methodological limitations of conventional neuro-imaging modalities”. Therefore, we designed this study to remove the effect of cortical plasticity due to both the duration and timing of implant use and the programming choices during the study [13]. To remove the above effects, we planned to measure the speech discrimination tests at the same time. 

The mean age at implantation was 55.2 years (range 23–79 years), and the time of implant use ranged from 1 to 24.4 years. Four types of speech processors were used (CP920, one patient; CP810, five patients; Freedom, three patients; Freedom for N22, two patients). The patient demographics and stimulation parameters are shown in Table 1. The default IIDR setting as an initial activation program at the switch-on session had been kept for at least six months in all patients.

### 2.2. IIDR Settings and Other Parameters before Speech Discrimination Tests

For IIDR selection, two settings (T-SPL/C-SPL of 25/65 dB (default IIDR; CSPL65: IIDR 40 dB) and T-SPL/C-SPL of 25/80 dB (wide IIDR; CSPL80: IIDR 55 dB)) with ADRO and no ASC were assigned simultaneously without displaying the setting configuration details to the patients (Figure 1). 

The C levels in the two IIDR settings were set to be the same in terms of loudness scaling at a conversational voice level across the IIDR settings without exceeding a comfortable level. The sensitivity setting was fixed at the default setting of 12, except for one subject setting it at 9 because of the patient’s preferred sound quality in the early initial programming session, and ADRO was applied for all patients. Each subject tried the two settings for at least three months and self-selected the one they preferred for sound quality at the end of the setting trial (initial IIDR selection). After initial IIDR selection, all subjects were provided with only their selected IIDR settings with the fixed settings of volume and sensitivity until the speech discrimination tests. Subjects were divided into two groups according to selected IIDR settings in the initial IIDR selection (default IIDR: preCSPL65 (*n* = 5); wide IIDR: preCSPL80 (*n* = 6)). Adaptation periods from the date of the initial IIDR selection to the date of speech discrimination testing ranged from 0.5 to 4.4 years.

### 2.3. Evaluation of Speech Discrimination Performance

All subjects underwent all word and sentence tests in one session on the same day at two different C-SPL settings (default and wide IIDR) without ASC. Both IIDR settings were evaluated using the CI-2004 recorded word and sentence test materials in quiet and noise, presented at conversational and loud speech levels (65/80 dB SPL) under stimulus calibration with speech-weighted unmodulated noise using a sound level meter (NL-42A; Rion Co., Tokyo, Japan). The test signals were presented using a single loudspeaker in front of each subject. The lists of CI-2004 word and sentence tests were changed in each test setting and IIDR condition to minimize potential learning and order effects. CI-2004 word and sentence tests are composed of either a 25-word test or a 15-sentence test, involving a total of 60 morphemes in each of the eight word/sentence test lists. The word test was scored on the total number of correct words and the sentence test was scored on the total number of correct morphemes. Noise tests were administered at a SNR of +10 dB with noise as speech-weighted unmodulated noise. All tests were performed using one word list and one sentence list per test setting and IIDR condition, respectively. Both IIDR settings were evaluated with recorded materials in quiet and noise presented at 65 and 80 dB SPL. All test conditions are described below.

IIDR setting; IIDR 40 dB (CSPL65: default IIDR)/IIDR 55 dB (CSPL80: wide IIDR)Test with presentation level of 65/80 dB SPL; words in quiet/words in noise (SNR = +10 dB)/sentences in quiet/sentences in noise (SNR = +10 dB).

### 2.4. Statistical Analyses

The word and sentence tests in CI-2004 did not follow complete identical distribution across the test lists because the variation in extracted words and morphemes in each word and sentence list was widely distributed to prevent practice effects in the test, and the accuracy rate for all consonants and vowels varied greatly for each subject. All possible dimensions in this study, including sex, participant age, duration of deafness, duration of cochlear implant use, type of speech processor, implant type, cording strategy, number of sensitivities, number of activated channels, number of maxima, and stimulation rate, could not be matched perfectly in both IIDR groups and could induce selection bias in various pairwise tests. Therefore, the Monte Carlo permutation test, a resampling statistical method that can completely evaluate multidimensional definite integrals with complicated boundary conditions, was performed with an R studio of 1.3.959 because of the advantages in sample complexity [14]. This test compares the changes in two observed score conditions, such as between a pre-score and a post-score. The expected distribution is generated by permuting the two conditions of an observed score under the assumption of a null hypothesis using computational repeated random sampling algorithms. If the result revealed that the null hypothesis was true, the two observed score conditions would not be significantly different. If the null hypothesis was rejected, the two observed score conditions would have a relative difference, and the *p*-value represents the significance.

The Monte Carlo permutation test was performed as follows. A random score sample (word test: 25 words, sentence test: 60 morphemes in 15 sentences) was permuted from scores with two different IIDR settings, and a leftover score sample was created. The mean scores of both the random score sample and the leftover score sample were calculated, and the mean difference between the two mean scores was measured 100,000 times without replacement resampling. After calculating the extreme value by summing the fraction of the random permuted mean differences equal to or greater than the actual mean differences between the pre- and post-scores, the *p*-values for the permutation test were calculated by dividing the extreme value by 100,000.

## 3. Results

### 3.1. CI-2004 Word and Sentence Tests at Conversational and Loud Speech Levels in All Subjects

The individual results of each test are shown in Figure 2. Individual results revealed that tests in noise and at presentation of 80 dB tended to decrease word and sentence test scores.

Figure 3A,B shows the word and sentence tests at a conversational speech level (65 dB SPL) and at a loud speech level (80 dB SPL) in quiet and in noisy conditions with default and wide IIDR in all subjects. The wide IIDR showed significantly higher scores compared with the default IIDR for words in quiet (*p* < 0.05), words in noise (*p* < 0.01), and sentences in noise (*p* < 0.01) at a conversational speech level and words in quiet (*p* < 0.05), words in noise (*p* < 0.01), sentences in quiet (*p* < 0.01), and sentences in noise (*p* < 0.01) at a loud speech level.

### 3.2. CI-2004 Word and Sentence Tests at Conversational and Loud Speech Levels in the preCSPL65 Group

Figure 3C,D shows the word and sentence tests at a conversational speech level (65 dB SPL) and at a loud speech level (80 dB SPL) in quiet and noisy conditions with default and wide IIDR in the preCSPL65 group. The wide IIDR showed significantly higher scores compared with the default IIDR for words in noise (*p* < 0.05) at a conversational speech level and words in noise (*p* < 0.01) and sentences in noise (*p* < 0.01) at a loud speech level.

### 3.3. CI-2004 Word and Sentence Tests at Conversational and Loud Speech Levels in the preCSPL80 Group

Figure 3E,F shows the word and sentence tests at a conversational speech level (65 dB SPL) and at a loud speech level (80 dB SPL) in quiet and noisy conditions with default and wide IIDR in the preCSPL80 group. The wide IIDR showed significantly higher scores compared to the default IIDR for words in quiet (*p* < 0.05), words in noise (*p* < 0.01), and sentences in noise (*p* < 0.01) at a conversational speech level and all test lists (*p* < 0.01) at a loud speech level.

### 3.4. CI-2004 Word and Sentence Tests at Conversational and Loud Speech Levels with Default/Wide IIDR in the preCSPL65 and preCSPL80 Group

Figure 4 shows the word and sentence tests at a conversational speech level (65 dB SPL) and at a loud speech level (80 dB SPL) in quiet and noisy conditions with default and wide IIDR in all subjects separated by initial IIDR selection (preCSPL65 or preCSPL80). The results showed both significant differences between preCSPL65 and preCSPL80 in all test lists (*p* < 0.01) and a clear separation in these groups when evaluating words and sentences in noise at a conversational speech level (Figure 4A,B). From Figure 4B, most of the preCSPL80 scores in words and sentences in noisy conditions showed a severe drop to approximately 0% at a loud speech level. On the other hand, with a wide IIDR, there was no significant difference in all tests at a loud speech level.

## 4. Discussion

The wide IIDR in this study theoretically covers 25 to 80 dB SPL by changing the envelope clipping threshold from 65 dB to 80 dB SPL. Our results show that a wide IIDR was significantly effective for speech perception of words in quiet, words in noise, and sentences in noise presented at 65 dB SPL as the normal conversation level. The mean scores for words in quiet conditions differed slightly between the IIDRs (default IIDR, 78.9%; wide IIDR, 85.8%). In noisy environments, a wide IIDR showed clear differences in score range. Additionally, a wide IIDR showed significant improvement in all conditions during the word and sentence tests presented at 80 dB SPL as the loud conversation level. The range of all scores clearly differed between the IIDRs. 

These results imply that a wide IIDR is effective in a noisy environment during normal conversations and in all conditions during loud conversations. The reasons for the effectiveness of a wide IIDR in these conditions are that a wide IIDR provides (1) a clear SNR in psychophysical electric stimulation, (2) better access to the envelope cues of speech in a noisy environment, and (3) clear envelope cues of speech without infinite envelope compression at the AGC threshold of 65 dB SPL.

Experimental studies comparing programming settings often led to controversial interpretations because of the limitations of adaptation periods to the parameter changes [4]. In this study, the groups were divided by initial IIDR settings considering adaptation, and the effectiveness of a wide IIDR was measured in the two groups. The preCSPL65 experiment showed that a wide IIDR did not improve speech perception scores, except for words in noise presented at 65 dB SPL, and the difference in the mean scores of words in noise was only 7.2% (default IIDR, 70.4; wide IIDR, 77.6), but a wide IIDR significantly improved the scores for words and sentences in noise presented at 80 dB SPL. In preCSPL80, all tests except sentences in quiet at 65 dB SPL showed significant differences between the default IIDR and wide IIDR.

These results suggest that a wide IIDR was effective in speech perception at a loud conversation level in noisy environments, and this effectiveness was not affected by the adaptation period. The other significant differences in preCSPL80 may result from subjects being accustomed to the IIDR setting; therefore, a long adaptation to a wide IIDR could improve speech perception in all conditions except normal conversations in a quiet environment.

A wide IIDR would have improved speech perception in all patients, but the initial IIDR selection was different due to subject preference. Our results revealed that the preCSPL65 group showed a better hearing score for normal conversations than the preCSPL80 group, and hearing scores in noisy environments were over 50% at the normal conversation level in default IIDR. The hearing scores of the preCSPL80 group in noisy environments were mostly under 50% at the normal conversational level and nearly 0% at the loud conversational level in default IIDR. These scores were completely separated, and there were no overlaps between the preCSPL65 and preCSPL80 groups.

Patients with severe hearing difficulty in noisy environments in default IIDR might prefer a wide IIDR setting for daily use, and patients showing good perception in noisy conditions in default IIDR might prefer default IIDR rather than wide IIDR. The loudness of soft to normal conversation in default IIDR is larger than that in wide IIDR because of the translation mechanism of the acoustic input level into the EDR via the IIDR setting [5]. Thus, patients selecting the initial default IIDR (preCSPL65) might emphasize the sense of loudness in soft to normal conversation over the difficulty of hearing loud conversations in a noisy environment.

In addition, some elderly candidates live in solitude, which does not allow for frequent noise exposure or opportunities to converse with other people. On the other hand, other elderly candidates live together with family, which provides opportunities to converse with loud voices. Wide IIDR is effective, especially in loud conversations in noisy environments, but the aforementioned types of lifestyles might be one of the factors in initial IIDR selection. Those who live alone in a quiet environment, such as residing in the countryside, might prefer the default IIDR setting for daily life, and those who talk to others who speak loudly and/or have frequent noise exposure, such as living with family in roadside houses, might prefer the wide IIDR setting.

Currently, Cochlear Ltd. devices after the launch of the Cochlear Nucleus 6 system (CP910/920 processor) have data logging functions that record various listening environments, device usage, and loudness levels. This function will help estimate the effectiveness of the wide IIDR settings. The preference for each IIDR differs by individual, but research involving a wide IIDR is worthwhile, especially for people who often have loud conversations and/or encounter noisy environments. A wide IIDR does not cause speech perception to decline in normal conversations; rather, it improves hearing ability in loud conversations and noisy environments.

The COVID-19 pandemic has led many people to impose a distance between listener and talker [15], and the use of face coverings such as conventional face masks often impedes using lipreading cues. Additionally, phone conversation and darkness also lead to hearing difficulty for the same reason. Lipreading is useful for people with hearing loss to help convey information when communicating in a noisy environment [16]. Lacking lipreading cues led to poor speech understanding and greater need for concentration in noisy environments, especially in middle-aged and older adults [17]. Therefore, wide IIDR, which leads to better perception in noisy conditions, would also be useful for CI patients, especially middle-aged and older adults, when communicating with people wearing face masks, during telephone conversations, and in darkness.

This study has some limitations. First, subjects recruited in this study were all unilateral CI patients because current Japanese conformity criteria for cochlear implants do not fully allow access to bilateral CI surgery. The criteria allow bilateral CI surgery only for exploring binaural hearing. Second, the effect of ASC was not measured because ASC settings were not commonly preferred in our previous CI patients for the reason of wavering loudness after loud sound exposure. ASC reduces the sensitivity of the speech processor microphone when the noise floor reaches or exceeds 57 dB SPL, and this reduction in sensitivity reduces the loudness of the sound and noise, which may affect hearing ability, especially in noisy conditions and loud speech levels. Third, the Monte-Carlo permutations test was applied for removing possible selection bias factors, such as age, type of speech processor, duration of CI use, as well as number of cases, etc. The number of cases was perfectly adequate for statistical analysis because of the power of Monte-Carlo methods, but other possible factors associated with higher brain function, including cortical plasticity, might also be correlated with the observed effects in each subject. By contrast, further analysis will be needed to analyze the factors of age differences and lifestyle conditions (silent vs. noisy) regarding this effectiveness, because our current study had an insufficient number of patients to analyze statistical differences in this aspect. However, the results may guide future studies on wide IIDR by estimating effectiveness, including the difference in adaptation periods as a confounding factor. Further research on recommended individual configuration settings is needed to improve hearing performance in any environment, but wide IIDR can benefit patients who often have loud conversations or are in noisy environments.

## 5. Conclusions

In conclusion, the present study shows the effectiveness of wide IIDR in improving speech perception during loud conversations in noisy environments without any adaptation period. A long adaptation period to wide IIDR also improves speech perception in all conditions, except normal conversations in a quiet environment. Wide IIDR may provide a new personalized setting method to improve hearing, especially in loud conversations and noisy environments.

## Figures and Tables

**Figure 1 jpm-12-01860-f001:**
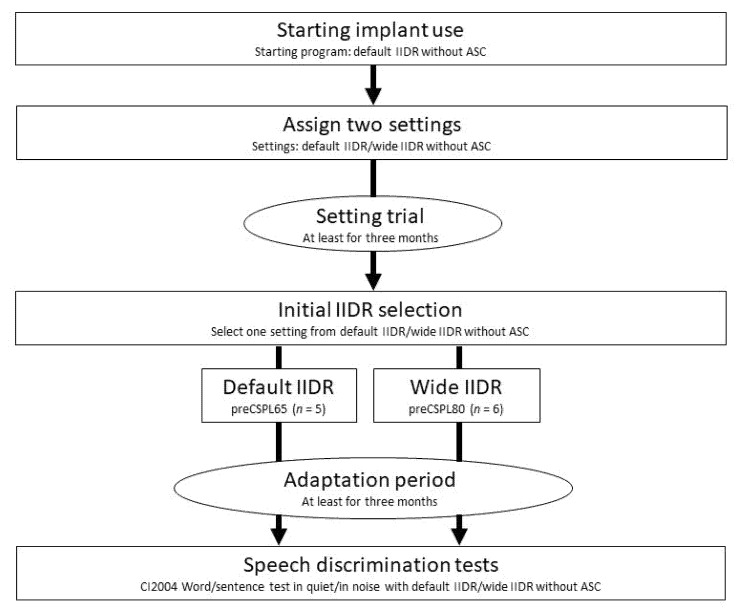
Summary of study procedures. The research protocol was initiated at the time of “assign two settings”.

**Figure 2 jpm-12-01860-f002:**
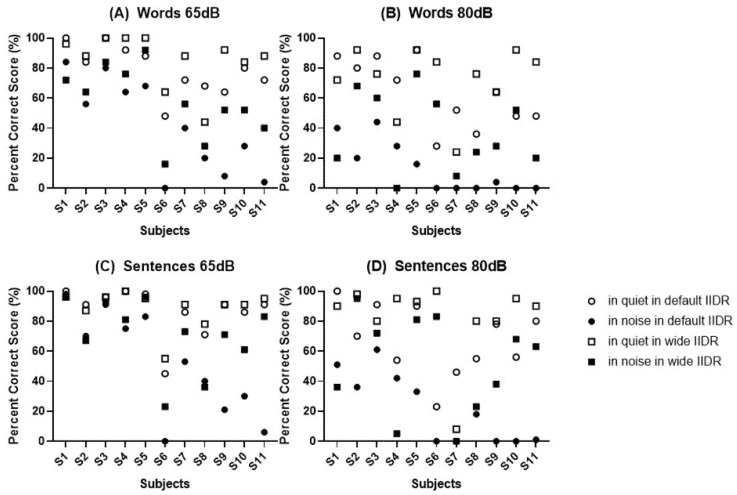
Individual scores for each test type and presentation level. (**A**) Word test at 65 dB, (**B**) word test at 80 dB, (**C**) sentence test at 65 dB, and (**D**) sentence test at 80 dB.

**Figure 3 jpm-12-01860-f003:**
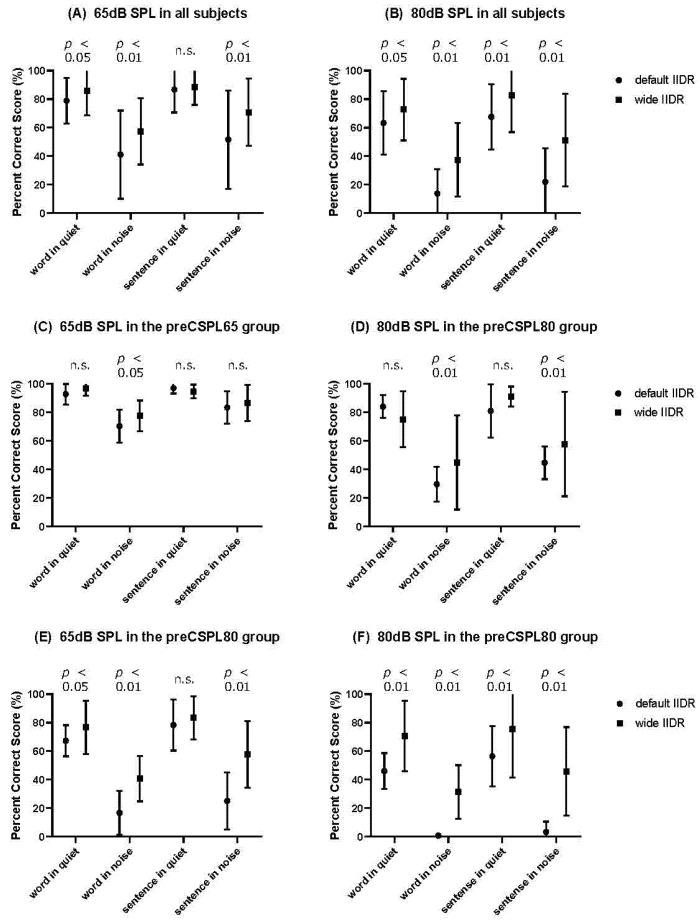
Percentage of correct word and sentence test scores in quiet and noise conditions presented at 65/80 dB SPL in (**A**,**B**) all subjects, (**C**,**D**) the preCSPL65 group, and (**E**,**F**) the preCSPL80 group. Mean scores of words in quiet, words in noise, sentences in quiet, and sentences in noise conditions were (**A**) 78.9%, 41.4%, 86.8%, 51.5%, (**B**) 63.3%, 13.8%, 67.5%, 22.0%, (**C**) 92.8%, 70.4%, 97.0%, 83.4%, (**D**) 84.0%, 29.6%, 81.0%, 44.6%, (**E**) 67.3%, 16.6%, 78.3%, 25.0%, and (**F**) 46.0%, 0.7%, 56.3%, 3.2%, respectively, with default IIDR, and (**A**) 85.8%, 57.5%, 88.6%, 70.9%, (**B**) 72.7%, 37.5%, 82.6%, 51.2%, (**C**) 96.8%, 77.6%, 94.8%, 86.6%, (**D**) 75.2%, 44.8%, 91.2%, 57.8% (**E**) 76.7%, 40.7%, 83.5%, 57.8%, and (**F**) 70.7%, 31.3%, 75.5%, 45.8%, respectively, with wide IIDR. Significant differences were seen in the conditions of words in quiet (*p* < 0.01; (**F**), *p* < 0.05; (**A**,**B**,**E**)), words in noise (*p* < 0.01; (**A**,**B**,**D**–**F**), *p* < 0.05; (**C**)), sentences in quiet (*p* < 0.01; (**B**,**F**)), and sentences in noise (*p* < 0.01; (**A**,**B**,**D**–**F**)).

**Figure 4 jpm-12-01860-f004:**
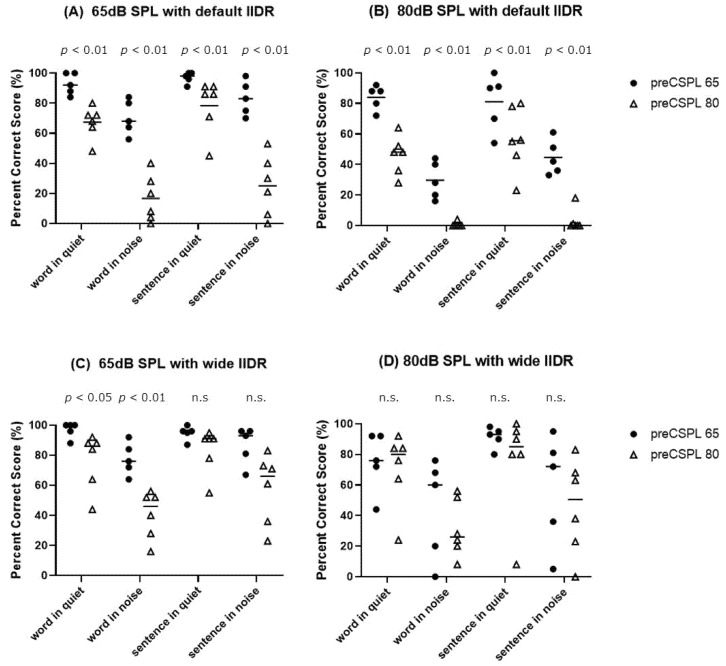
CI2004 word and sentence test results with default IIDR/wide IIDR compared between the preCSPL65 and preCSPL80 groups at (**A**,**C**) normal and (**B**,**D**) loud conversation levels. Mean words in quiet, words in noise, sentences in quiet, and sentences in noise scores were (**A**) 92.8%, 70.4%, 97.0%, 83.4%, (**B**) 84.0%, 29.6%, 81.0%, 44.6%, (**C**) 96.8%, 77.6%, 94.8%, 86.6%, and (**D**) 75.2%, 44.8%, 91.2%, 57.8%, respectively, in preCSPL65 and (**A**) 67.3%, 16.6%, 78.3%, 25.0%, (**B**) 46.0%, 0.67%, 56.3%, 3.2%, (**C**) 76.7%, 40.7%, 83.5%, 57.8%, and (**D**) 70.7%, 31.3%, 75.5%, 45.8%, respectively, in preCSPL80. (**A**,**B**) Significant differences were seen in all tests (*p* < 0.01), and there was a clear separation between words in noise and sentences in noise. (**C**) Significant differences were seen in quiet (*p* < 0.05) and in noise (*p* < 0.01) conditions. (**D**) There was no significant difference in all tests.

**Table 1 jpm-12-01860-t001:** Details of subjects.

ID	Sex	Age (yr)	Etiology of Sensorineural Hearing Loss	Duration of Deafness (yr)	Implant Use (yr)	Speech Processor	Implant Type	Coding Strategy	Sensitivity	Number of Channels	Number of Maxima	Rate	Regular C-SPL Setting (dB)	Experience Duration of Wide IIDR as Regular Program (yr)
S1	F	60	Unknown	0.8	1.9	CP810	CI422(SRA)	ACE	12	22	9	3500	65	NA
S2	F	60	Unknown	1	1.7	CP810	CI24RE(CA)	ACE	12	22	12	1200	65	NA
S3	M	37	Unknown	0.5	3.3	CP810	CI24RE(CA)	ACE	12	21	12	1200	65	NA
S4	F	35	Usher syndrome	4.8	13	CP810	CI24M	ACE	12	22	16	900	65	NA
S5	M	24	Large vestibular aquaduct syndrome	0.9	5.7	Freedom	CI24RE(CA)	ACE	12	18	12	1200	65	NA
S6	F	67	Unknown	13.4	6.1	Freedom	CI24R(CS)	ACE	12	20	12	1200	80	4.4
S7	F	80	Unknown	14.6	1	CP920	CI24RE(CA)	ACE	12	22	12	1200	80	0.5
S8	M	67	Unknown	0.4	1.6	CP810	CI24RE(CA)	ACE	12	22	12	1200	80	0.7
S9	F	64	Unknown	9.4	5	Freedom	CI24RE(CA)	ACE	12	22	11	900	80	4.0
S10	F	78	Unknown	3	15	Freedom22	CI22M	SPEAK	9	22	10	233–300	80	1.5
S11	F	42	meningitis	5.4	24.4	Freedom22	CI22M	SPEAK	12	18	9	210–256	80	0.7

## Data Availability

Not applicable.

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
