# Peer review of "Effectiveness of Expanded Instantaneous Input Dynamic Range Adjustment on Speech Perception"

_jpm, 2022, doi:10.3390/jpm12111860_

Round 1

Reviewer 1 Report (Previous Reviewer 3)

As highlighted in the text, further studies are possible. I would suggest differentiating the ages, checking the differences between adults and children, also taking into account the lifestyle (silent vs noisy).

Author Response

As highlighted in the text, further studies are possible. I would suggest differentiating the ages, checking the differences between adults and children, also taking into account the lifestyle (silent vs noisy).

Thank you for your reviewing our manuscript.

We also wanted to analyze the effectiveness in the differences of the ages and the lifestyle, but the methodological difficulties caused the limitation of further step in this manuscript. To analyze in this point, we think that it is necessary to obtain more patient number to get statistical difference in the effect, and we estimate it to be more than 80 patients.

Therefore, we address about it in the manuscript.

Reviewer 2 Report (New Reviewer)

This is an interesting study on the differences between default and wide IIDR usage, also at conversational level, and also at loud levels. The article is suitable for publication, however minor changes are recommended

Page 3 line 108-116: the authors discuss the effect of brain plasticity in the first 3 years post-implantation. Table 1 describes the amount of years post implantation, and we can see some were less than 3 years and some longer. This is not a factor mentioned later on in the results, so should be emitted or further explained.

Table 1 contains 2 columns that are the same: regular C-SPL and group of IIDR

Figure 1: please explain which staage is the background, and  when was the research protocol initiated

Page 5: no information on ethics committee included in the results section, this should be added

Figure 3: the % in the text are unclear, also heading of figure D should be preCSPL65 group

Figure 4:  should include the opposite of using wide IIDR and comparing between the preCSPL65 and preCSPL80 group

Page 10 lines 332-339: It is unclear why they refer to COVID-19 and masks, the same is relevant to phone conversations, dark  etc.

Author Response

This is an interesting study on the differences between default and wide IIDR usage, also at conversational level, and also at loud levels. The article is suitable for publication, however minor changes are recommended

>Thank you for your reviewing our manuscript. We responded your suggestions answering below mentioned.

Page 3 line 108-116: the authors discuss the effect of brain plasticity in the first 3 years post-implantation. Table 1 describes the amount of years post implantation, and we can see some were less than 3 years and some longer. This is not a factor mentioned later on in the results, so should be emitted or further explained.

>Thank you for giving us the recommendation. To make it clear, we add a sentence in the paragraph.

Table 1 contains 2 columns that are the same: regular C-SPL and group of IIDR

>Thank you for giving us the recommendation. To make it clear, we removed a column of group of IIDR.

Figure 1: please explain which stage is the background, and when was the research protocol initiated

>Thank you for giving us the recommendation. To make it clear, we add the detail in the figure legend.

Page 5: no information on ethics committee included in the results section, this should be added.

>Thank you for giving us the suggestion. We already mentioned about it in the section like “Institutional Review Board Statement: This study was approved by the ethics committee of the Hiroshima University Faculty of Medicine (E-1788)”.

Figure 3: the % in the text are unclear, also heading of figure D should be preCSPL65 group

>Thank you for giving us the suggestion. We changed the figure 3 in terms of the suggestion.

Figure 4:  should include the opposite of using wide IIDR and comparing between the preCSPL65 and preCSPL80 group

>Thank you for giving us the suggestion. We changed the figure 4 in terms of the suggestion.

Page 10 lines 332-339: It is unclear why they refer to COVID-19 and masks, the same is relevant to phone conversations, dark  etc.

>Thank you for giving us the suggestion. We changed the paragraph adding the above situation.

This manuscript is a resubmission of an earlier submission. The following is a list of the peer review reports and author responses from that submission.

Round 1

Reviewer 1 Report

Abstract

Just looking at the abstract I have a problem with the study design. You say you randomly assigned the two settings but then you let patients decide which one they prefer and you go from there... That is not randomly assigning....

And then you say that after the trial the two settings were tested (speech perception). That means that one is tested acutely and for the other one the patient had months to adjust. That is just not a fair comparison.

Introduction

NH subjects can hear more than 100 dB. They typically can hear 130-140dB but that is the pain threshold

Neurelec does not exist anymore. Now and for the time being until they officially become Cochlear, it's Oticon Medical.

Methods

The problem with using such a wide range of implant models, when you have such a small sample is that every implant system AND sound processor will handle the dynamic range a bit different so its like comparing apples and oranges...

Reviewer 2 Report

Page 2, lines 84-86, 87-88; Need references.

Page 5, lines 151-152: Any specific reason why authors tested at +10 dB SNR?

Page 5, lines 160-163: Did the authors mean that the actual test CI-2004 test materials are not normally distributed or the data from this study? This has to be clarified.

Method

A lot of key information is missing in the methods section. Stimulus calibration? Distance between speaker and the participant. How long was the test duration?

Results

Figure 3: Description should be reduced.

Discussion

Lines 338-346: I feel this paragraph is not relevant.

Reviewer 3 Report

The work is well written and suitable for publication. It provides numerous food for thought. However, the number of patients considered is small.

My personal experience is that IDR and IIDR are highly dependent on the patient and his/her lifestyle. Before the numerical perceptual results, the well-being of the patient must be considered. A patient can benefit more from using an implant, even if he feels numerically less well.

A large IDR / IIDR allows the perception of a more normal and complete sound, which also includes ambient noises. There are people who ask to reduce ambient noise because it causes them annoyance and reduce their QoL. Other people prefer to hear all sounds, even the weakest ones.

It is obvious that in a whispered or normal conversation, the presence of ambient sounds can have the effect of a white noise, masking the perception of the voice.

Much also depends on the location of the primary sound source and secondary sound sources (ambient sounds) relative to the implant.

In further study I suggest the authors to consider all these aspects, or test them separately.